

# Association between the metabolic score for insulin resistance and prostate cancer: a cross-sectional study in Xinjiang

Jinru Wang[1,*], Aireti Apizi[2,*], Ning Tao[1] and Hengqing An[2]

[1] College of Public Health, Xinjiang Medical University, Urumqi, Xinjiang Uygur Autonomous Region, China
[2] Department of Urology, The First Affiliated Hospital of Xinjiang Medical University, Urumqi, Xinjiang Uygur Autonomous Region, China
* These authors contributed equally to this work.

## ABSTRACT

**Background:** Insulin resistance is associated with the development and progression of various cancers. However, the epidemiological evidence for the association between insulin resistance and prostate cancer is still limited.

**Objectives:** To investigate the associations between insulin resistance and prostate cancer prevalence.

**Methods:** A total of 451 patients who were pathologically diagnosed with prostate cancer in the First Affiliated Hospital of Xinjiang Medical University were selected as the case population; 1,863 participants who conducted physical examinations during the same period were selected as the control population. The metabolic score for insulin resistance (METS-IR) was calculated as a substitute indicator for evaluating insulin resistance. The Chi-square test and Mann-Whitney U test were performed to compare the basic information of the case population and control population. Univariate and multivariate logistic regression analyses to define factors that may influence prostate cancer prevalence. The generalized additive model (GAM) was applied to fit the relationship between METS-IR and prostate cancer. Interaction tests based on generalized additive model (GAM) and contour plots were also carried out to analyze the interaction effect of each factor with METS-IR on prostate cancer.

**Results:** METS-IR as both a continuous and categorical variable suggested that METS-IR was negatively associated with prostate cancer prevalence. Smoothed curves fitted by generalized additive model (GAM) displayed a nonlinear correlation between METS-IR and prostate cancer prevalence ($P < 0.001$), and presented that METS-IR was negatively associated with the odds ratio (OR) of prostate cancer. The interaction based on the generalized additive model (GAM) revealed that METS-IR interacted with low-density lipoprotein cholesterol (LDL-c) to influence the prostate cancer prevalence ($P = 0.004$). Contour plots showed that the highest prevalence probability of prostate cancer was achieved when METS-IR was minimal and low-density lipoprotein cholesterol (LDL-c) or total cholesterol (TC) was maximal.

**Conclusions:** METS-IR is nonlinearly and negatively associated with the prevalence of prostate cancer. The interaction between METS-IR and low-density lipoprotein cholesterol (LDL-c) has an impact on the prevalence of prostate cancer. The study suggests that the causal relationship between insulin resistance and prostate cancer still needs more research to confirm.

Corresponding authors
Ning Tao, 38518412@qq.com
Hengqing An, 9269735@qq.com

# INTRODUCTION

Prostate cancer is a serious disease that threatens male health, its rapid progression, ease of metastasis, and invisibility pose major challenges for prostate cancer screening and treatment. The global pandemic of metabolic diseases has become an urgent public health problem to be solved (*Guerra et al., 2021*). Insulin resistance is a characteristic of metabolic disorders (*Yaribeygi et al., 2018*) and is defined as impaired insulin absorption and glucose utilization (*Ormazabal et al., 2018*). When insulin sensitivity is reduced, it is usually accompanied by insulin resistance. Insulin resistance is manifested by reduced sensitivity of muscle, liver, and adipose tissue to insulin stimulation, causing glucose metabolism disorder and leading to compensatory insulin elevation and hyperinsulinemia.

Several epidemiologic studies have suggested that insulin resistance is associated with an increased risk of breast cancer, thyroid cancer, liver cancer, endometrial cancer, and pancreatic cancer (*Giovannucci et al., 2010*; *Iwase et al., 2021*; *Yin et al., 2018*). However, existing findings on the relationship between insulin resistance and prostate cancer have been inconsistent. *Jochems et al. (2023)* showed that insulin resistance markers are not associated with the risk of clinically relevant prostate cancer. *Fritz et al. (2024)* applied triglyceride-glucose (TyG) index as an indicator of insulin resistance, which is not associated with prostate cancer death in the entire cohort. *Häggström et al.*'s *(2018)* cohort study found no link between type 2 diabetes mellitus and prostate cancer risk. *Christakoudi et al. (2024)* suggested that obesity, diabetes and prostate cancer risk are synergistically negatively correlated, possibly involving reduced testosterone in diabetic patients and oestrogen production in obese patients. *Peila & Rohan (2020)* and *Wang, Yang & Liao (2020)* obtained a significant negative association between diabetes and prostate cancer risk. *Monroy-Iglesias et al.*'s *(2021)* study showed that the metabolic syndrome is not associated with prostate cancer risk, and no correlation is found between triglycerides, high-density lipoprotein cholesterol (HDL-c), blood pressure, waist circumference and prostate cancer risk, but glycated hemoglobin (HbA1c) is consistently negatively correlated with prostate cancer risk. After excluding other possible confounding factors, *Gao et al. (2022)* found that metabolic syndrome, hypertension, hyperlipidaemia, hyperglycaemia and obesity are not associated with prostate cancer. However, *Hernández-Pérez et al. (2022)* proposed that metabolic syndrome and some of its components as potential risk factors for prostate cancer. The association between insulin resistance and prostate cancer is currently uncertain for the following reasons: (1) The definition and criteria of metabolic syndrome are different in diverse countries (*Bhindi et al., 2015*; *Tande, Platz & Folsom, 2006*). (2) There are disparities in the level of economic development in different countries or regions, leading to different rates of prostate cancer screening (*Bhindi et al., 2015*). (3) The prevalence of prostate cancer varies between different ethnicities of patients with

metabolic syndrome (*Beebe-Dimmer et al., 2009*). Therefore, we reduce the bias through the following methods: (1) The quantitative indicator METS-IR was taken to evaluate the relationship between metabolic syndrome or insulin resistance and prostate cancer, avoiding the use of different definitions or criteria for metabolic syndrome. (2) This study employed data from a large general hospital in Xinjiang was able to reduce the influence of economic level differences in different regions on prostate cancer screening. (3) Xinjiang is located in northwestern of China, at the crossroads of Central and West Asia, and is a region of multi-ethnic coexistence, which avoids the bias that studying a particular ethnic group bring to the results. In the Xinjiang region, various ethnic groups develop different dietary habits as and lifestyle (*Li et al., 2019*), distinct from the inland provinces of China. Studies have found that special dietary patterns and lifestyles in the Xinjiang region influence the prevalence of metabolic syndrome and its components (*Zhang et al., 2024*; *Hailili et al., 2021*; *Wang et al., 2021*). Meanwhile, the prevalence of prostate cancer shows ethnic and regional differences (*Zeng et al., 2023*; *Delon et al., 2022*). Compared with other regions, the prevalence of metabolic syndrome and prostate cancer varies to some extent in the Xinjiang region. The euglycemic-hyperinsulinemic clamp (EHC) is currently recognized as the gold standard for determining insulin sensitivity in humans (*Rebelos & Honka, 2020*). Regrettably, euglycemic-hyperinsulinemic clamp (EHC) is invasive and expensive, time and labour consuming, and its clinical application is limited, which cause it to be difficult to apply on a large scale in the population. Fasting insulin-based indicators of insulin resistance, such as the homeostasis model assessment (HOMA), quantitative insulin sensitivity check index (QUICKI), also have limitations in terms of accuracy and stability (*Muniyappa et al., 2008*). The development of non-insulin-based indicators of insulin resistance has provided a simpler method of detecting insulin resistance, particularly in primary care settings. The metabolic insulin resistance score (METS-IR) is a newly developed non-insulin-based indicator of insulin resistance. Compared to other insulin-based or non-insulin-based indicators, METS-IR has stronger correlation with euglycemic-hyperinsulinemic clamp (EHC) and METS-IR is more accurate for euglycemic-hyperinsulinemic clamp (EHC) (*Bello-Chavolla et al., 2018*). METS-IR has high diagnostic performance for reduced insulin sensitivity, with an AUC of 0.845 (95% CI: [0.783–0.899]) for recognizing insulin resistance (*Bello-Chavolla et al., 2018*). METS-IR integrated non-insulin fasting experimental and anthropometric values, body composition is an important component of nutrition and metabolism (*Brown, Harhay & Harhay, 2019*). Therefore, METS-IR is more comprehensive than previous insulin resistance indicators and can be easily obtained in physical examination. METS-IR in this study was used as an indicator to evaluate insulin resistance, and higher METS-IR indicates greater insulin resistance. The relationship of insulin resistance and prostate cancer was measured by studying the correlation between METS-IR and prostate cancer. Therefore, we conducted this study based on prostate cancer patients in Xinjiang region, used the quantitative variable METS-IR as an indicator to explain the association between metabolic syndrome or insulin resistance and prostate cancer.

## MATERIALS AND METHODS

### Study population

This study was a population-based cross-sectional study, conducted by medical record review, all patients and participants admitted to the hospital sign an informed consent form. We extracted patient information from the Department of Urology of the First Affiliated Hospital of Xinjiang Medical University in 2023, patients diagnosed histologically for the first time with prostate cancer by prostate biopsy. The duration of prostate cancer was from the first diagnosis of prostate cancer to 2023. Participants who underwent physical examination during the same period as the control population.

Inclusion criteria for control population and case population: (1) In the control population, males attending for the physical examination in the same period, and age more than 50 years; In the case population, patients first diagnosed with prostate cancer by prostate biopsy; (2) complete test data; (3) can read, understand and sign the informed consent form.

Exclusion criteria for control population and case population: (1) In the control population, participants with any type of cancer or history of cancer; in the case population, prostate cancer patients with a history of other types of cancer; (2) participants with a history of diabetes mellitus and the use of glucose-lowering drugs; (3) participants with a history of diseases related to lipid metabolism disorders, such as liver or kidney disease, and the use of triglyceride-lowering drugs.

There were 3,494 patients in the Department of Urology of the First Affiliated Hospital of Xinjiang Medical University, and 1,474 of them were prostate cancer patients, 2,020 patients with other urological cancers were not included in the study. According to exclusion criteria, 451 prostate cancer patients were included in the study as the case population. The control population was from 12,262 male participants who underwent physical examination during the same period in the First Affiliated Hospital of Xinjiang Medical University, and after excluding those who did not meet the criteria, 1,863 participants were included in the study as the control population. See Fig. 1.

## METHODS

The study was reviewed and approved by the Ethics Committee of First Affiliated Hospital of Xinjiang Medical University (approval number: 20220308-166).

### Physical examination, laboratory tests and pathological diagnosis

The basic information, physical examination, and laboratory samples of the subjects were collected by professional doctors or nurses in the hospital. All participants were dressed lightly, the shoes were removed. The height was measured to the nearest centimetre by using a rangefinder, and weight was measured to the nearest 0.1 kg by using a body composition analyser.

Participants fasted overnight for 10–12 h on the first day of admission, venous blood specimens were collected from the anterior elbow vein at 8:00–9:00 am on the second day of admission, and sent to the haematology department. Total cholesterol (TC), fasting triglycerides (TG), high-density lipoprotein cholesterol (HDL-c), low-density lipoprotein
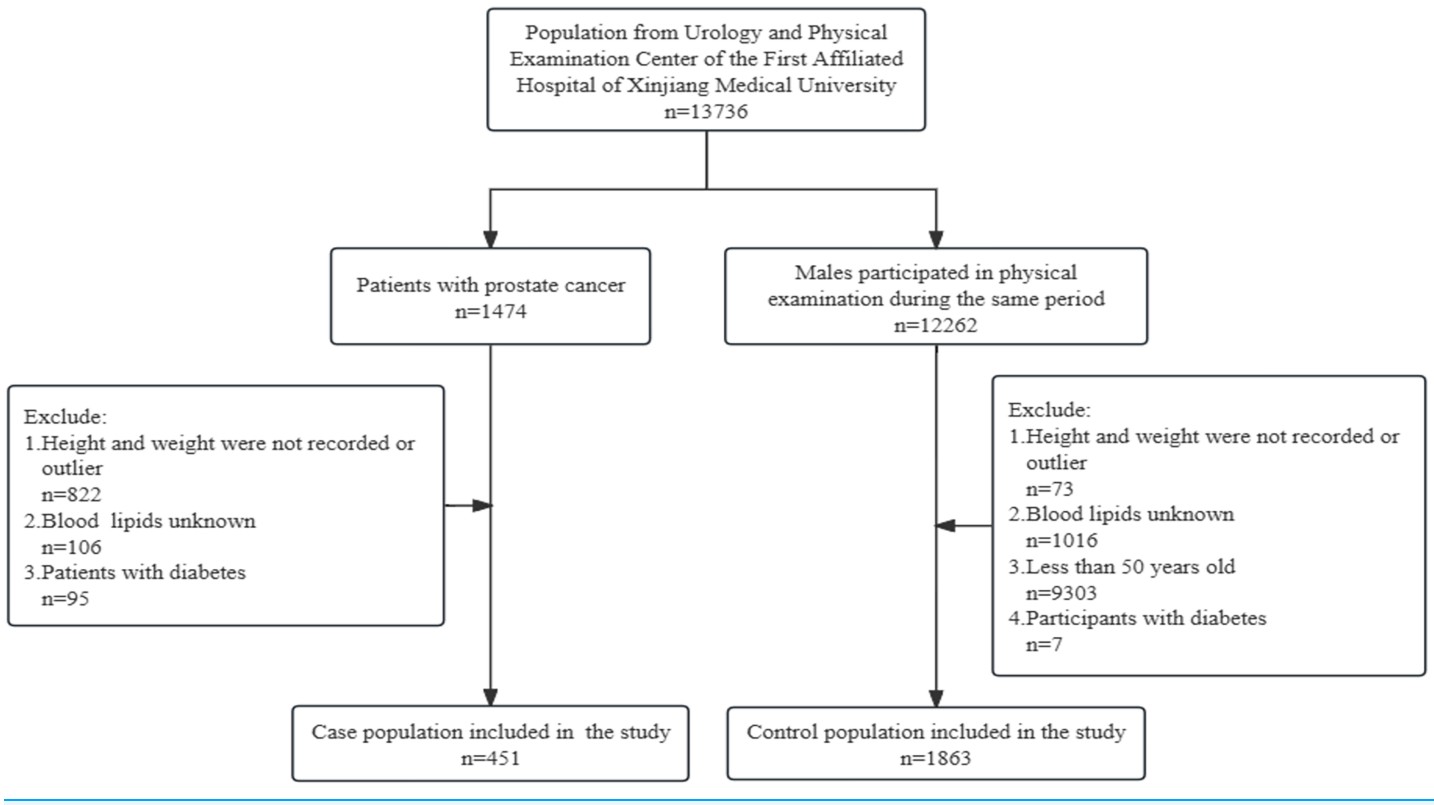

**Figure 1 Flowchart of the sample selection.**

cholesterol (LDL-c), fasting glucose, and alkaline phosphatase were detected by automatic biochemical analyzer.

## Indicator definition

Body mass index (BMI) was defined as weight/height$^2$ (Kg/m$^2$). METS-IR was calculated as an indicator to evaluate the severity of insulin resistance, METS-IR = (Ln((2* G0) + TG0)* BMI)/(Ln(HDL-c)) (G0: fasting glucose (mg/dL), TG0: fasting triglycerides (mg/dL), BMI: body mass index (Kg/m$^2$), HDL-c: high-density lipoprotein cholesterol (mg/dL)) (*Bello-Chavolla et al., 2018*).

## Statistical methods

SPSS (version 26.0) and R (version 4.3.2) software were applied to analyze the data. Based on the entire population, METS-IR was divided into quartiles. Q1 group was less than 35.240, Q2 group was 35.240–39.605, Q3 group was 39.605–44.410, Q4 group was greater than 44.410. Skewed distribution data were expressed as M (P$_{25}$, P$_{75}$), normal distribution data were expressed as $\bar{\chi}$ ± s, count data were expressed as case (%). The basic information was compared within the case and control populations using Mann-Whitney U test and $\chi^2$ test. We constructed three logistic regression analysis models to analyze the relationship between METS-IR and prostate cancer. Model 1 did not adjust for any covariates; Model 2 was adjusted for age based on Model 1; Based on Model 2, Model 3 adjusted for hypertension, total cholesterol (TC), low-density lipoprotein cholesterol (LDL-c), blood

calcium (Ca), and alkaline phosphatase. A generalized additive model (GAM) was constructed to describe the nonlinear relationship between METS-IR and prostate cancer. In succession, we performed an interaction test based on the generalized additive model (GAM), plotted the interaction model in three dimensions. In the three-dimensional diagram, the higher the relative Z-axis value, the greater the interaction effect of the two variables on prostate cancer. We verified the stability of the interaction results by plotting contour plots. Different types of colors in the contour plots represent different probabilities of prostate cancer prevalence. The interaction models were used to explore whether there was an interaction between blood calcium (Ca), total cholesterol (TC), low-density lipoprotein cholesterol (LDL-c), and METS-IR; $P < 0.05$ was considered statistically significant.

## RESULTS

### Basic information of study population

A total of 2,314 participants were included in the study, with 451 prostate cancer patients in the case population and 1,863 participants in the control population. Contrasting alkaline phosphatase in the case and control populations, the difference was not statistically significant ($P > 0.05$). In the case and control populations, fasting glucose were statistically significant ($P < 0.05$). Comparing the age, prevalence of hypertension, body mass index (BMI), fasting triglycerides (TG), total cholesterol (TC), high-density lipoprotein cholesterol (HDL-c), low-density lipoprotein cholesterol (LDL-c), blood potassium (K), and blood calcium (Ca) in the case and control populations, the differences were statistically significant ($P < 0.001$). For our indicator of interest, METS-IR (continuous) and METS-IR (quartile), there were statistically significant differences between the case and control populations. See Table 1.

### Relationship between METS-IR and prostate cancer prevalence
*Univariate and multivariate logistic regression analysis*

The odds ratios (ORs) of METS-IR were consistently significant in all three models irrespective of whether METS-IR was analyzed as a continuous variable or quartiles. When METS-IR was evaluated as a continuous variable, in the all variables adjusted model (model 3), the adjusted OR was 0.942 (95% CI: [0.921–0.964]). When METS-IR was analyzed as quartiles, also in model 3, the adjusted OR in Q2, Q3, and Q4 were 0.579 (95% CI: [0.382–0.879]), 0.488 (95% CI: [0.320–0.743]), and 0.407 (95% CI: [0.264–0.629]), respectively, with Q1 as reference. See Table 2.

### Smooth curve of the generalized additive model

The smoothed curve of the generalized additive model (GAM) was constructed to evaluate the relationship between METS-IR and prostate cancer. Generalized additive models (GAM) displayed a nonlinear correlation between METS-IR and the prevalence of prostate cancer ($P < 0.001$). The smooth curve of the generalized additive model (GAM) presented that METS-IR was negatively associated with the odds ratio (OR) of prostate cancer. See Fig. 2.
**Table 1 Basic information about the subjects included in the study.**

| Characteristic | Case population ($n = 451$) | Control population ($n = 1,863$) | $\chi^2$/Z | P |
|---|---|---|---|---|
| Age (years)/(cases (%)) | | | 693.274 | <0.001 |
| ≤54 | 16 (3.50%) | 1,335 (71.70%) | | |
| >54 | 435 (96.50%) | 528 (28.30%) | | |
| Hypertension/(cases (%)) | | | 291.494 | <0.001 |
| Yes | 319 (70.70%) | 1,791 (96.10%) | | |
| No | 132 (29.30%) | 72 (3.90%) | | |
| *Body mass index (Kg/m$^2$)/(M(P$_{25}$, P$_{75}$)) | 24.09 (22.00–26.45) | 25.82 (23.95–27.99) | −9.925 | <0.001 |
| *Fasting glucose (mg/dL)/(M(P$_{25}$, P$_{75}$)) | 96.59 (85.23–115.33) | 97.67 (90.10–113.71) | −2.944 | 0.003 |
| *Fasting triglycerides (mg/dL)/(M(P$_{25}$, P$_{75}$)) | 118.72 (88.60–160.37) | 142.65 (98.35–207.32) | −6.556 | <0.001 |
| Total cholesterol (mmol/L)/(M(P$_{25}$, P$_{75}$)) | 4.13 (3.49–4.81) | 5.02 (4.39–5.70) | −16.037 | <0.001 |
| *High-density lipoprotein cholesterol (mg/dL)/(M(P$_{25}$, P$_{75}$)) | 40.22 (32.48–49.11) | 46.40 (40.60–53.75) | −10.103 | <0.001 |
| Low-density lipoprotein cholesterol (mmol/L)/(M(P$_{25}$, P$_{75}$)) | 2.60 (2.05–3.21) | 3.07 (2.57–3.67) | −10.721 | <0.001 |
| Blood potassium (mmol/L)/(M(P$_{25}$, P$_{75}$)) | 3.79 (3.55–4.11) | 4.17 (3.95–4.41) | −15.892 | <0.001 |
| Blood calcium (mmol/L)/(M(P$_{25}$, P$_{75}$)) | 2.23 (2.14–2.32) | 2.35 (2.28–2.42) | −16.827 | <0.001 |
| Alkaline phosphatase (U/L)/(M(P$_{25}$, P$_{75}$)) | 74.00 (61.20–102.20) | 76.00 (65.00–90.00) | −0.817 | 0.414 |
| *Metabolic score for insulin resistance (METS-IR)/(M(P$_{25}$, P$_{75}$)) | 38.28 (33.62–43.25) | 39.91 (35.60–44.66) | −5.255 | <0.001 |
| *Metabolic score for insulin resistance (METS-IR)/(cases (%)) | | | 27.678 | <0.001 |
| Q1 (<35.240) | 155 (34.40%) | 424 (22.80%) | | |
| Q2 (35.240–39.605) | 107 (23.70%) | 471 (25.30%) | | |
| Q3 (39.605–44.410) | 99 (22.00%) | 480 (25.80%) | | |
| Q4 (>44.410) | 90 (20.00%) | 488 (26.20%) | | |

Notes:
* Variables used to calculate metabolic score for insulin resistance (METS-IR).
METS-IR was an indicator of research focus.
METS-IR = (Ln ((2 * G0) + TG0) * BMI)/(Ln (HDL-c)) (G0: fasting glucose (mg/dL); TG0, fasting triglycerides (mg/dL); BMI, body mass index (Kg/m$^2$); HDL-c, high-density lipopro-tein cholesterol (mg/dL)).

**Table 2 Univariate and multivariate logistic regression analysis.**

| METS-IR | Model 1 | | Model 2 | | Model 3 | |
|---|---|---|---|---|---|---|
| | P | OR (95% CI) | P | OR (95% CI) | P | OR (95% CI) |
| Continuous | <0.001 | 0.955 [0.940–0.970] | <0.001 | 0.959 [0.942–0.977] | <0.001 | 0.942 [0.921–0.964] |
| Categories | | | | | | |
| Q1 (<35.240) | Reference | | Reference | | Reference | |
| Q2 (35.240–39.605) | 0.001 | 0.621 [0.470–0.822] | 0.008 | 0.631 [0.450–0.886] | 0.010 | 0.579 [0.382–0.879] |
| Q3 (39.605–44.410) | <0.001 | 0.564 [0.425–0.749] | 0.001 | 0.561 [0.399–0.790] | 0.001 | 0.488 [0.320–0.743] |
| Q4 (>44.410) | <0.001 | 0.504 [0.377–0.675] | 0.001 | 0.550 [0.388–0.781] | <0.001 | 0.407 [0.264–0.629] |
| P for trend | <0.001 | | <0.001 | | <0.001 | |

Notes:
Model 1: Crude; unadjusted model.
Model 2: Adjusted for age.
Model 3: Additionally, adjusted for hypertension, total cholesterol, low-density lipoprotein cholesterol, blood calcium, and alkaline phosphatase.

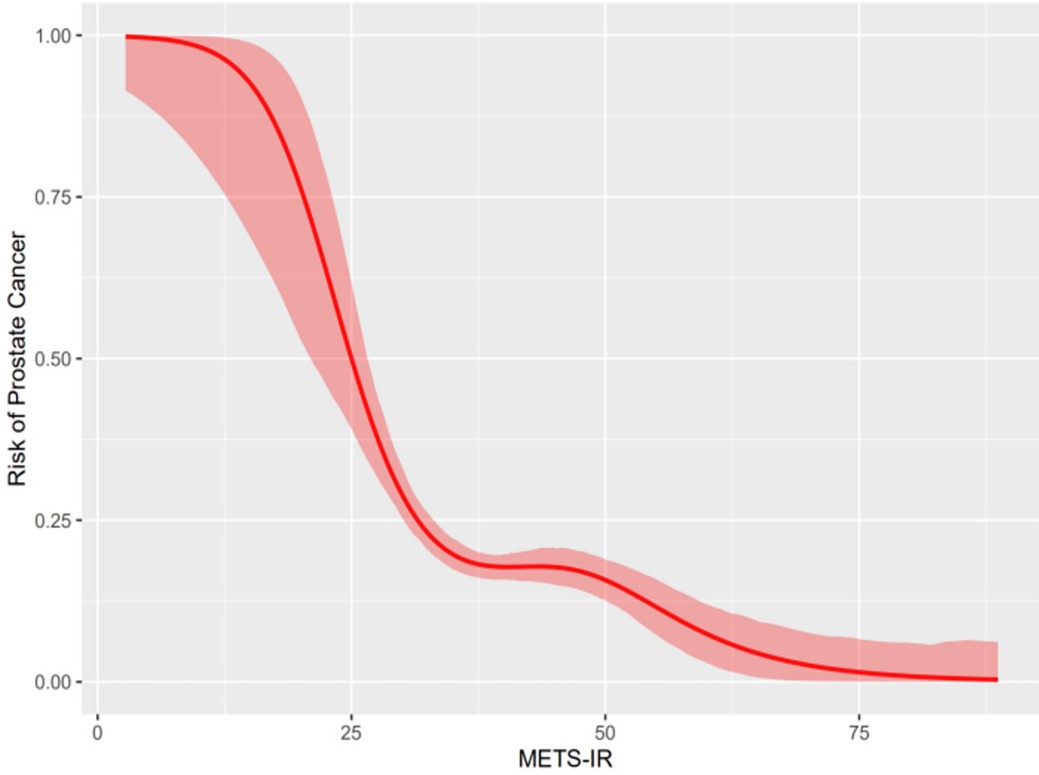

**Figure 2** Relationship between METS-IR and prostate cancer.

## Interaction of METS-IR with low-density lipoprotein cholesterol, blood calcium, and total cholesterol on prostate cancer

### Interaction tests on the basis of generalized additive model

We constructed interaction tests on the basis of generalized additive model (GAM) to assess the interaction effect of METS-IR with different variables on prostate cancer. There was an interaction effect between METS-IR and low-density lipoprotein cholesterol (LDL-c) on prostate cancer ($\chi^2$ = 15.03, $P$ = 0.004). METS-IR < 20 and low-density lipoprotein cholesterol (LDL-c) >8 mmol/L were associated with the highest odds ratio (OR) of prostate cancer; when METS-IR > 20 and low-density lipoprotein cholesterol (LDL-c) <8 mmol/L, the odds ratio (OR) of prostate cancer gradually decreased. See Fig. 3.

The results suggested that the interaction between METS-IR and blood calcium (Ca) was not statistically significant on prostate cancer ($\chi^2$ = 0.897, $P$ = 0.443). See Fig. S1. The interaction effect between METS-IR and total cholesterol (TC) on prostate cancer was not statistically significant ($\chi^2$ = 8.471, $P$ = 0.081). See Fig. S2.

### Interaction tests of contour plot

To further validate the specific interaction effects between METS-IR and different variables on prostate cancer, contour plots were plotted to display the probability of prostate cancer prevalence. The different types of colors in the contour plot represent different prevalence probabilities of prostate cancer. The highest prevalence probability of prostate cancer was
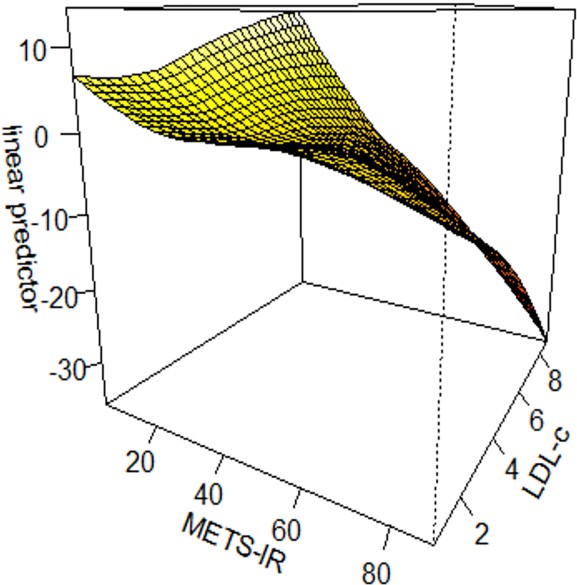

**Figure 3 Interaction test on the basis of GAM between METS-IR and low-density lipoprotein cholesterol on prostate cancer.**

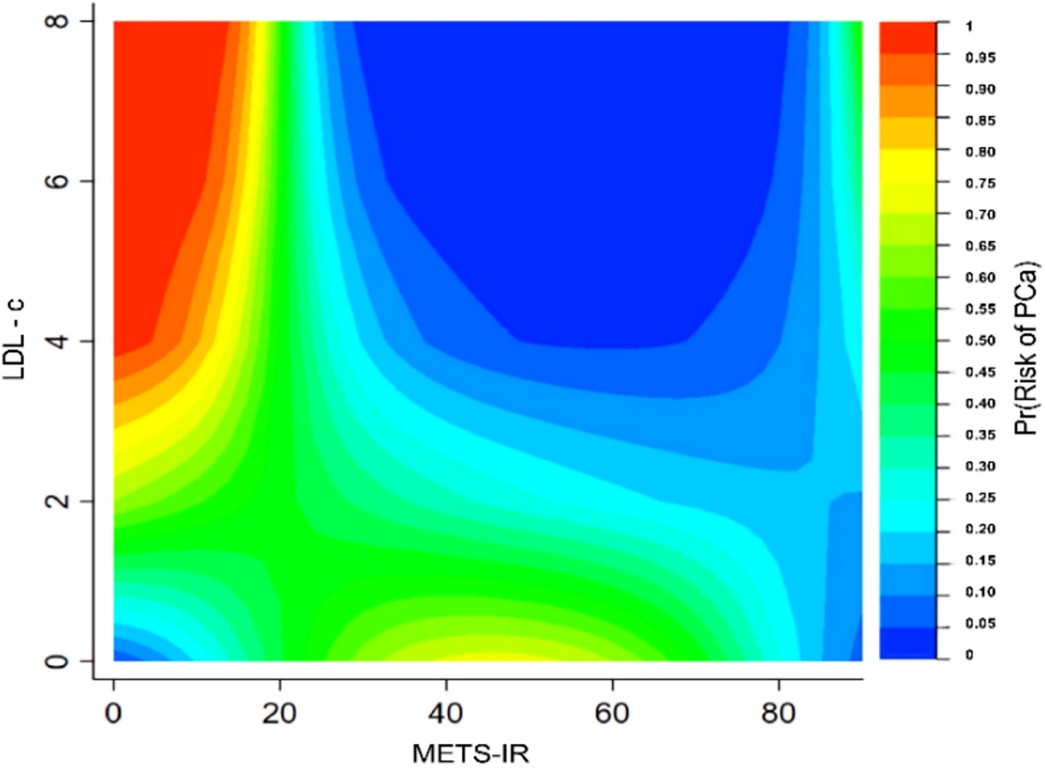

**Figure 4 Interaction test of contour plot between METS-IR and low-density lipoprotein cholesterol on the probability of prostate cancer prevalence.**

achieved when METS-IR was minimal and low-density lipoprotein cholesterol (LDL-c) or total cholesterol (TC) was maximal. See Figs. 4 and S3.

## DISCUSSION

Prostate cancer is a heterogeneous disease with a complex etiology, previous studies that evaluated the association between the degree of diabetes or insulin resistance and the risk of prostate cancer have shown conflicting results. According to a meta-analysis of 29 cohort studies and 16 case-control studies, involving 8.1 million participants and 132,331 prostate cancer patients, type 2 diabetes mellitus was associated with reduced risk of prostate cancer (*Bansal et al., 2013*; *Peila & Rohan, 2020*). A Mendelian randomization study found no association between glycemic characteristics, type 2 diabetes and prostate cancer risk (*Au Yeung & Schooling, 2019*). Diagnostic markers of insulin resistance also have lacked correlation with prostate cancer in a cohort study (*Jochems et al., 2023*). It indicated that more sensitive indicators are needed to assess the association between insulin resistance and prostate cancer. METS-IR as a novel surrogate indicator for evaluating insulin resistance, has a high sensitivity to identify insulin resistance cases in the general population, obese people, and type 2 diabetes mellitus patients (*Bello-Chavolla et al., 2018*). Besides, it also contains variables such as blood lipids and body composition, which are more comprehensive than previous substitute indicators. All variables used to calculate METS-IR are mandatory items for physical examination, so it is promising that METS-IR can be applied on a large scale in the population as a substitute indicator of insulin resistance. Xinjiang is located in the hinterland of Eurasia, various ethnic groups live in large, small communities, and there are differences in diet structure, lifestyle, and genetic background. The disease characteristics of prostate cancer are closely related to racial, genetic and other factors (*Siegel et al., 2020*), the epidemiological characteristics of prostate cancer in Xinjiang region have its uniqueness. Therefore, the current study investigates the relationship between METS-IR and prostate cancer based on the Xinjiang population.

The essence of insulin resistance is decreased insulin sensitivity and insulin responsiveness. Insulin is produced by the β-cells of the pancreas, which promotes the absorption of glucose and fatty acids as well as the synthesis of fats and proteins, while inhibiting the breakdown of fats and proteins. The pancreatic β-cell dysfunction is central to the pathophysiological mechanisms of insulin resistance. When glucose levels are elevated, insulin secretion by pancreatic β-cells is stimulated. Notable is that insulin resistance is not entirely pathological, insulin resistance also exists under normal physiological conditions, such as those who are in adolescence, high mental stress, sedentary work, and intake of high-calorie foods (*White, Shaw & Taylor, 2016*; *Stefanaki et al., 2020*).

Insulin resistance or diabetes mellitus is associated with a lower risk of prostate cancer, however, the mechanism has not been clarified. *Christakoudi et al. (2024)* suggested that obesity and diabetes are synergistically negatively associated with prostate cancer risk, possibly related to reduced testosterone in diabetes or insulin-resistant patients, and diminished oestrogen production in obesity (*Grossmann, 2011*). In a collaborative analysis

of 20 prospective studies, *Watts et al. (2018)* researched circulating levels of testosterone and prostate cancer risk, and found that males with low testosterone have a lower risk of suffering from prostate cancer. Androgen is a known driver of prostate cancer, the development of prostate cancer is dependent on androgen and androgen receptors. Testosterone is one of the major activators of androgen receptor (*Lutz et al., 2018*), lower testosterone predicts lower risk of prostate cancer. In addition to androgen, androgen receptor activity in prostate cancer is regulated by cholesterol derivatives, steroid hormones can antagonise androgen signalling through the oestrogen receptor and other pathways (*Krycer & Brown, 2011*). Cholesterol is a precursor substance for the synthesis of steroid hormones, 27-hydroxycholesterol as the most oxidized derivative of plasma cholesterol can deplete cholesterol in cells and inhibit prostate cancer cells growth (*Alfaqih et al., 2017*).

Numerous scholars have investigated the association between blood lipids and the prostate cancer prevalence. However, most of these studies were observational and did not provide definitive conclusions (*Michalakis et al., 2015*; *Murtola et al., 2018*; *Ioannidou et al., 2022*; *Lutz et al., 2022*). Insulin resistance and dyslipidemia may be mutually causal (*Bjornstad & Eckel, 2018*), we found an interaction between insulin resistance and low-density lipoprotein cholesterol (LDL-c) in prostate cancer patients. Insulin resistance plays an important role in the development of dyslipidemia. In a 32-month intervention and follow-up study, high levels of oxidized low-density lipoprotein cholesterol (LDL-c) coexist with insulin resistance (*Linna et al., 2015*), and elevated low-density lipoprotein cholesterol (LDL-c) associated with increased prostate cancer risk (*Bull et al., 2016*).

In the conditions of overnutrition, hyperglycaemia and hyperlipidaemia, can lead to pancreatic islet inflammation. During the development of obesity, large numbers of macrophages from the bone marrow accumulate in islets, which causes pancreatic islet inflammation, leading to pancreatic islet β-cell dysfunction (*Sica & Mantovani, 2012*; *Sivakumar et al., 2021*). Overnutrition plays an important role in insulin resistance. Excessive energy intake drives insulin resistance and metabolic disease progression (*Gancheva et al., 2018*). The personal fat threshold hypothesis suggests that excessive ectopic fat accumulation in susceptible individuals may lead to insulin resistance and hyperglycaemia (*Janssen, 2024*). In situations of overnutrition, excessive fat intake will inevitably increase fat oxidation when fat storage reserves are saturated, and by inducing competition for oxidative substrates, aggravating insulin resistance (*Bray & Bouchard, 2020*; *Girousse et al., 2018*). A follow-up study discovered that insulin resistance increased with increased energy intake, suggesting that insulin resistance may be a protective response in favour of the intracellular environment (*Buscemi et al., 2024*). Study including 259,884 men from eight European cohorts, 11,760 of who developed prostate cancer, indicated that there was a negative correlation between insulin resistance and prostate cancer incidence (*Fritz et al., 2024*). Most studies often use body mass index (BMI) as a measurable criterion for obesity and body composition. Patients with early or advanced tumors may lose weight due to poor appetite or increased metabolic demands (*Arem et al., 2013*). Insulin resistance appears to be mediated by subcutaneous or peripheral fat deposition sites (*Cheung et al., 2016*), and higher body mass index (BMI) was associated

with a lower risk of prostate cancer (OR = 0.91, 95% CI [0.85–0.98]) (*Loh et al., 2022*). A meta-analysis of epidemiological studies reported that for every 5-unit increase in body mass index (BMI), there was a six percent reduction in the risk of localized prostate cancer (*Kyrgiou et al., 2017*). Another potential driver of the negative association between obesity and prostate cancer development may be hyperleptinemia, mediation analyses indicated that mediators may account for up to 50 percent of the protective effect of genetically determined body mass index (BMI) on prostate cancer risk (*Loh et al., 2022*). Meanwhile, hyperglycemia and tumor necrosis factor-α (TNF-α) decrease androgen receptors through nuclear factor kappa B (NF-kB) (*Barbosa-Desongles et al., 2013*), thereby reducing the risk of prostate cancer. Some studies have shown that elevated glycosylated hemoglobin (HbA1c) is negatively associated with the risk of developing prostate cancer, due to lower concentrations of insulin-like growth factor-1 (IGF-1) in men with higher glycosylated hemoglobin (HbA1c) (*Monroy-Iglesias et al., 2021*). Prospective study demonstrated that insulin-like growth factor-1 (IGF-1) is positively associated with the risk of developing prostate cancer (*Roddam et al., 2008*; *Watts et al., 2020*). The use of insulin is also associated with reduced cancer risk. *Yang et al. (2010)* found that there is a negative correlation between exposure to exogenous insulin and cancer risk, and the correlation between the two is consistent no matter how the queue is analysed. In a diabetic state, α-cells usually secreted higher than normal glucagon levels (*Asadi & Dhanvantari, 2021*), glucagon promotes the synthesis of insulin-like growth factor binding protein-1 (IGFBP-1), insulin-like growth factor binding protein-1 (IGFBP-1) partly reduces the activity of insulin-like growth factor (IGF) (*McCarty, 1997*), thus reducing the cancer risk.

In the study, we employed the generalized additive model (GAM) to explore the non-linear relationship between METS-IR and prostate cancer prevalence. The advantages of the generalized additive model (GAM) consist of two main elements: one hand, compared to traditional linear models, the generalized additive model (GAM) can fit complex non-linear relationships between variables; on the other hand, compared to machine learning, the generalized additive model (GAM) can contribute significance levels and confidence intervals for the regression coefficients, enabling the model results more interpretable. The association between METS-IR and prostate cancer is influenced by multiple factors, and the effects of these factors on prostate cancer are sometimes shown to be synergistic or antagonistic. Interaction tests based on generalized additive model (GAM) and contour plots can be used to explore and verify if these factors interact with METS-IR. Undeniably, our study also has some limitations. First, as a cross-sectional study, we were unable to establish a causal relationship between altered METS-IR and prostate cancer, longitudinal studies with larger samples should be conducted, to explore the temporal sequence of insulin resistance and prostate cancer occurrence. Second, although we adjusted for confounders that may have interfered with the study results, there are still some other uncontrolled factors. Third, we did not assay HOMA-IR, which has been more widely used in past research to assess insulin resistance.

## CONCLUSIONS

In conclusion, we demonstrated METS-IR as an emerging surrogate indicator of insulin resistance, has a non-linear and negative correlation with the prostate cancer prevalence. Concurrently, there is an interaction between METS-IR and low-density lipoprotein cholesterol (LDL-c) on prostate cancer prevalence. The complexity of the causes of prostate cancer requires multidisciplinary cross-fertilization to prevent and treat prostate cancer.

### Funding

This work was supported by the Excellence Youth Science Foundation of Xinjiang Uygur Autonomous Region (grant number 2023D01E05); the Key Projects of Xinjiang Uyghur Autonomous Region (grant number 2022D01D39); the National Natural Science Foundation of China (grant number 82360476); the Xinjiang Uygur Autonomous Region "Tianshan Talents" youth science and technology top talent project (grant number 2022TSYCCX0097). The funders had no role in study design, data collection and analysis, decision to publish, or preparation of the manuscript.

### Grant Disclosures

The following grant information was disclosed by the authors:
Excellence Youth Science Foundation of Xinjiang Uygur Autonomous Region: 2023D01E05.
Key Projects of Xinjiang Uyghur Autonomous Region: 2022D01D39.
National Natural Science Foundation of China: 82360476.
Xinjiang Uygur Autonomous Region "Tianshan Talents" youth science and technology top talent project: 2022TSYCCX0097.

### Competing Interests

The authors declare that they have no competing interests.

### Author Contributions

- Jinru Wang conceived and designed the experiments, performed the experiments, analyzed the data, prepared figures and/or tables, and approved the final draft.
- Aireti Apizi conceived and designed the experiments, analyzed the data, prepared figures and/or tables, and approved the final draft.
- Ning Tao conceived and designed the experiments, analyzed the data, prepared figures and/or tables, authored or reviewed drafts of the article, and approved the final draft.
- Hengqing An conceived and designed the experiments, analyzed the data, prepared figures and/or tables, authored or reviewed drafts of the article, and approved the final draft.

## Human Ethics

The following information was supplied relating to ethical approvals (*i.e.*, approving body and any reference numbers):

Ethics Committee of First Affiliated Hospital of Xinjiang Medical University (Ethical Application: 20220308-166)

## Data Availability

The raw measurements are available in the Supplemental File.

## Supplemental Information

Supplemental information for this article can be found online at http://dx.doi.org/10.7717/peerj.17827#supplemental-information.

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
