# Peer review of "Association between the metabolic score for insulin resistance and prostate cancer: a cross-sectional study in Xinjiang"

_PeerJ, doi:10.7717/peerj.17827_

## Round 0.1 · original submission · Major Revisions

Dear Dr. Wang,

We have received two reviewer reports on your manuscript entitled "Association between the metabolic score for insulin resistance and the risk of prostate cancer". The reviewers have raised serious concerns regarding the overall methodological aspects and the lack of in-depth information necessary to support the study's conclusions.

To proceed with a second round of peer review, we advise you to respond to all the reviewer comments in a point-by-point manner. This will help justify the rationale and conclusions of your study more comprehensively.

Best regards,

Abhishek Tyagi
Associate Editor

Reviewer 1 ·

Basic reporting

This manuscript investigates the association between insulin resistance and prostate cancer prevalence. Although the study seems to be interesting, it was not possible to make a more thorough review because some pivotal information about the methods were lacking in the manuscript. I suggest the authors rewrite the manuscript including more information about the methods. For example, the text should cover the inclusion and exclusion criteria for subjects’ recruitment, elucidate whether the study was conducted by medical record review or patient recruitment, the time point of blood collection (admission, before treatment, after beginning of treatment), etc. Regarding the metabolic score for insulin resistance (METS-IR), it is not clear what is the reference range on a healthy patient, what increases or reductions from this range could suggest and if it is expected any shifts according to age. The sample seems not to be evenly distributed in the groups considering the age reported in table 1. It is also not possible to understand the criteria used in the quartile division methodology and multivariate logistic regression analysis. Ultimately, the authors should carefully proof-read spell check the manuscript to eliminate errors, long and confusing sentences, truncated information.

Experimental design

See comment above.

Validity of the findings

See comment above.

Additional comments

See comment above.

·

Basic reporting

1. The authors should provide more explicit clarification when stating, "However, the association between insulin resistance and prostate cancer is currently uncertain." This statement should be accompanied by a citation supporting their argument and a summary of the existing evidence regarding insulin resistance and prostate cancer, explaining why it is uncertain. Are there inconsistencies in the results of studies? Or is there a lack of evidence regarding this association?

Experimental design

2. One methodological aspect needing clarification is the study design. Based on the flow diagram and description, it appears to be more of a cross-sectional study in the methods section. A case-control study is unlikely due to the method of patient selection. Additionally, since the population under study should consist solely of men, "Being female" cannot be an exclusion criterion, as the event under consideration is prostate cancer. Instead, the authors should clarify the number of men affiliated with the Hospital of Xinjiang Medical University who had prostate cancer and who did not. It would also be crucial to mention the duration of prostate cancer in the study population.
3. Out of the total number of individuals meeting eligibility criteria, how many chose to participate in the study? Were there any differences between participants and non-participants? This information is crucial for assessing potential selection bias that may affect the validity of the study.
4. Figures 2 and 3 do not provide relevant information to the study, primarily because they display the coefficients of each variable in the regression model. The differences in estimators, such as for hypertension, may result from the variable METS-IR being included as continuous in one model and as quartiles in the other. Moreover, interpreting the coefficients of each variable may lead to potentially incorrect conclusions. For instance, there is evidence that calcium levels may affect blood pressure or cause hypertension, and hypertension itself has been independently associated with prostate cancer. Therefore, the coefficient for calcium is being adjusted for a variable (hypertension) that may mediate its association with prostate cancer, leading to overfitting. This possibly explains why the results suggest that higher calcium levels are associated with a lower prevalence of prostate cancer, contrary to existing evidence. Thus, I suggest consolidating the information into a single figure, focusing solely on the coefficients of METS-IR, and discontinuing the presentation of data as in Table 3.
5. The results presented in the supplementary material seem to exceed the scope of this study. Therefore, it would be better to mention in the introduction that there might be a synergistic or antagonistic effect of insulin resistance with other metabolic factors such as hypertension. In other words, the authors should mention that the metabolic syndrome is being evaluated in relation to prostate cancer because such a combined effect may exist. However, the authors do not provide a biological justification for this analysis.
6. In the discussion section, the authors should consider mentioning that it is unlikely for higher insulin resistance to result in lower prostate cancer risk. It is probably a consequence of reverse causality, and the way exposure was measured. This point is not addressed in the discussion. Additionally, the presence of selection bias cannot be ruled out since there is no information on the participation rate in this study.

Validity of the findings

2. One methodological aspect needing clarification is the study design. Based on the flow diagram and description, it appears to be more of a cross-sectional study in the methods section. A case-control study is unlikely due to the method of patient selection. Additionally, since the population under study should consist solely of men, "Being female" cannot be an exclusion criterion, as the event under consideration is prostate cancer. Instead, the authors should clarify the number of men affiliated with the Hospital of Xinjiang Medical University who had prostate cancer and who did not. It would also be crucial to mention the duration of prostate cancer in the study population.
3. Out of the total number of individuals meeting eligibility criteria, how many chose to participate in the study? Were there any differences between participants and non-participants? This information is crucial for assessing potential selection bias that may affect the validity of the study.
4. Figures 2 and 3 do not provide relevant information to the study, primarily because they display the coefficients of each variable in the regression model. The differences in estimators, such as for hypertension, may result from the variable METS-IR being included as continuous in one model and as quartiles in the other. Moreover, interpreting the coefficients of each variable may lead to potentially incorrect conclusions. For instance, there is evidence that calcium levels may affect blood pressure or cause hypertension, and hypertension itself has been independently associated with prostate cancer. Therefore, the coefficient for calcium is being adjusted for a variable (hypertension) that may mediate its association with prostate cancer, leading to overfitting. This possibly explains why the results suggest that higher calcium levels are associated with a lower prevalence of prostate cancer, contrary to existing evidence. Thus, I suggest consolidating the information into a single figure, focusing solely on the coefficients of METS-IR, and discontinuing the presentation of data as in Table 3.
5. The results presented in the supplementary material seem to exceed the scope of this study. Therefore, it would be better to mention in the introduction that there might be a synergistic or antagonistic effect of insulin resistance with other metabolic factors such as hypertension. In other words, the authors should mention that the metabolic syndrome is being evaluated in relation to prostate cancer because such a combined effect may exist. However, the authors do not provide a biological justification for this analysis.
6. In the discussion section, the authors should consider mentioning that it is unlikely for higher insulin resistance to result in lower prostate cancer risk. It is probably a consequence of reverse causality, and the way exposure was measured. This point is not addressed in the discussion. Additionally, the presence of selection bias cannot be ruled out since there is no information on the participation rate in this study.

Additional comments

Minor changes:
1. I suggest removing the word "risk" from the title, the study design does not permit the evaluation of prostate cancer risk.
2. In line with the above point, the introduction should detail how insulin resistance was evaluated in relation to prostate cancer, thus introducing or justifying the use of METS-IR. The methodological validity aspects should be moved to the discussion section as a potential strength of the study.
3. The flowchart refers to the selection of "Patients with urinary cancer," which should instead be "Patients with prostate cancer."
4. From lines 117-120, the inclusion of covariates in the different regression models is mentioned. However, I consider that this information should be included in the statistical analysis section.
5. The way of measure alkaline phosphatase is not mentioned.
6. Regardless of the statistical analysis section mentioning this, the table footnote of Table 1 should include the statistical tests used to compare the characteristics of interest between individuals with and without cancer.
7. In the methodology, it should be indicated that METS-IR was divided into quartiles and specify whether this was based on the entire population or on subjects without cancer.
8. In line 111, mentioning the suggestion of the statistical test is not relevant. Here, it is essential to be precise about the observed differences between patients with and without cancer.

---

## Round 0.2 · Minor Revisions

Dear Dr. Wang,

Thank you for your submission to PeerJ.

It is my opinion as the Academic Editor for your article - Association between the metabolic score for insulin resistance and prostate cancer: a cross-sectional study in Xinjiang - that it requires a few Minor Revisions as per the comments of R2.

Thanks

Abhishek Tygai, PhD
Academic Editor

·

Basic reporting

1. Clarity on Insulin Resistance and Prostate Cancer Association:
My last comment: “The authors should provide more explicit clarification when stating, "However, the association between insulin resistance and prostate cancer is currently uncertain." This statement should be accompanied by a citation supporting their argument and a summary of the existing evidence regarding insulin resistance and prostate cancer, explaining why it is uncertain. Are there inconsistencies in the results of studies? Or is there a lack of evidence regarding this association?”Author’s response: “...We explain the issue "the association between insulin resistance and prostate cancer is currently uncertain" in detail in the introduction…”
I appreciate the addition of this information in the introduction. However, I suggest summarizing this section (lines 59-93) and clearly delineating the gap in the literature regarding this association and how this study contributes. For instance, one limitation of previous studies on the association between metabolic syndrome components and prostate cancer is the variability in criteria used to identify these conditions. Does this study improve upon previous approaches in evaluating insulin resistance?
2. Study Population Criteria
In the study population section (lines 112-126), the inclusion and exclusion criteria for cases and controls could be collapsed. For example: both groups included participants with any type of cancer or history of cancer; (2) participants with a history of diabetes mellitus and the use of glucose-lowering drugs; (3) participants with a history of diseases related to lipid metabolism disorders, such as liver or kidney disease, and the use of triglyceride-lowering drugs. included participants without any history of cancer.
3. Relevance of Prostate Biopsy Details:
The detailed information about prostate biopsy as a diagnostic method is excessive and may not be relevant. It is enough to mention in the study population section: "“ We extracted patient information from the Department of Urology of the First Affiliated Hospital of Xinjiang Medical University in 2023, patients diagnosed histologically for the first time with prostate cancer by prostate biopsy

5. Presentation of Results:
Table 2 and Figure 2 appear to present overlapping information. I suggest keeping only the table or the figure.

Experimental design

.

Validity of the findings

4. Addressing Potential Confounding Factors:
It appears that obesity plays a crucial role (line: 275-286-, line 298-314) in elucidating because insulin resistance might be associated with a lower frequency of prostate cancer. Obesity could act as a confounder and an effect modifier, but the logistic regression models were not adjusted by obesity (lines 171-175): “We constructed three logistic regression analysis models to analyze the relationship between METS-IR and prostate cancer. Model 1 did not adjust for any covariates; Model 2 was adjusted for age based on Model 1; Based on Model 2, Model 3 adjusted for hypertension, total cholesterol (TC), low-density lipoprotein cholesterol (LDL-c), blood calcium (Ca), and alkaline phosphatase”
I suggest adjusting the models by obesity and possibly conducting a sensitivity analysis to evaluate the association across different levels of obesity. Furthermore, in considering whether overnutrition plays a role in this association (lines 297-306), adjustment by energy intake may be necessary. However, in the absence of such data, this limitation should be addressed in the discussion."

---

## Round 0.3 · accepted · Accept

Dear Dr. Wang,
Thank you for your submission to PeerJ.
I am writing to inform you that your manuscript - Association between the metabolic score for insulin resistance and prostate cancer: a cross-sectional study in Xinjiang - has been Accepted for publication.

Congratulations!

Warm regards,
Abhishek Tyagi
Academic Editor
PeerJ Life & Environment

·

Basic reporting

NA

Experimental design

NA

Validity of the findings

NA

Additional comments

NA